# Neural Tree Transducers for Tree to Tree Learning

## Abstract

We introduce a novel approach to tree-to-tree learning, the neural tree transducer (NTT), a top-down depth first context-sensitive tree decoder, which is paired with recursive neural encoders. Our method works purely on tree-to-tree manipulations rather than sequence-to-tree or tree-to-sequence and is able to encode and decode multiple depth trees. We compare our method to sequence-to-sequence models applied to serializations of the trees and show that our method outperforms previous methods for tree-to-tree transduction.

## 1 Introduction

The sequence-to-sequence (Seq2Seq) model has led to a revolution in generative models for neural encoding and decoding (Sutskever et al., 2014). Seq2Seq allowed for the encoding or compression of the initial input stream and a generative transformation sequence. From another point of view, Seq2Seq implicitly incorporate tree structure, whereas our method explicitly operates on trees. This is specifically important for tree transduction. We present a tree-to-tree (Tree2Tree) model for neural tree transduction which "translates" one tree into another. Intuitively, Seq2Seq can be thought of as equivalent to Seq2Tree, followed by Tree2Tree, and finally Tree2Seq. While these can be mathematically equivalent, for neural networks the structure can radically change what the model is able to learn.

Tree2Tree operates on trees as tree transducer. Tree transduction and tree rule extraction have been extensively studied outside of deep learning; for a review see Comon et al. (2007). Unlike classical transduction methods, neural Tree2Tree models have no explicit rule extraction. Tree2Tree extends neural implementation of tree structured transduction from the seminal work of Frasconi et al. (1998). We effectively learn the transduction rules instead of learning the weights on hand coded transduction rules. Tree transduction is an important problem for many tasks including neural syntactic machine translation (Cowan, 2008; Razmara, 2011; Wang et al., 2007), mapping sentences to functional programs (Alvarez-Melis & Jaakkola, 2017), and specialized program translation (Alur & D'Antoni, 2012).

Within natural language processing, Seq2Seq has been used for many applications including machine translation (Sutskever et al., 2014; Bahdanau et al., 2014; Vinyals et al., 2015b), caption generation (Vinyals et al., 2015b), abstractive summarization (Rush et al., 2015), dialog generation (Vinyals & Le, 2015). One of the main advantages of Seq2Seq other existing generative models like hidden Markov models is its ability to "remember" longer term information using long-short term memory (LSTM), or more recently gated recurrent units (GRUs). These nonlinear recurrent models have an explicit memory retention which allows for the information in the encoded sentence to persist over the generated sequence (Lin & Tegmark, 2016), furthermore, LSTM has been shown to capture context sensitive languages (Gers & Schmidhuber, 2001) and RNNs in general are Turing complete (Siegelmann & Sontag, 1995). Furthermore, Seq2Seq has been shown to learn grammar (Vinyals et al., 2015a) as well as tree structure (Bowman et al., 2015), and with *explicit supervision* syntax dependencies (Linzen et al., 2016). Nonetheless, the original Seq2Seq model from Sutskever et al. (2014) reversed the input sequence in order to increase performance, showing that memory retention of distant information remained a problem. Attentional mechanisms (Bahdanau et al., 2014; Luong et al., 2015) were introduced to assist the decoder to reference distant entries in the input sequence. There has been further use of memory storage additions (Sukhbaatar et al., 2015) to retain more relevant history.

Both sequence-to-tree (Aharoni & Goldberg, 2017; Zhang et al., 2015; Dong & Lapata, 2016) and tree-to-sequence (Eriguchi et al., 2016a) have been explored. However, tree-to-tree encoding and

decoding has been largely ignored. This, in spite of the fact that tree transduction and tree grammars have a long tradition (Engelfriet, 1975; Graehl & Knight, 2004; Cowan, 2008). Syntax based (tree-to-tree) machine translation models produced state-of-the-art results for many years (Cowan, 2008; Razmara, 2011).

Unlike other work that focuses on extending sequence-to-sequence to trees, our work frames the tree-to-tree problem in the context of tree transduction. We draw from a large body of previous work in this field and thus, explicitly examine this problem from the perspective of production rules and tree edit distance (Tai, 1979).

The major contribution of this paper is to introduce a neural tree transducer containing a new decoder mechanism specialized to tree-to-tree decoding and to show that this tree-to-tree model achieves better performance than standard Seq2Seq models on a clean set of transduction tasks.

## 2 TREE-TO-TREE MODEL

Our neural tree transducer uses an encoded representation of the input tree $S$ together with the generative decoder model described below to predict the output tree $T$. The tree $T$ is conditional on $S$ such that the conditional probability, $P(T \mid S) = \prod_{p \in Path(T)} P(x^p \mid x^1, \dots, x^{p-1}, \text{enc}(S))$ where $Path(T)$ is the sequence of integers indexing the tree traversal, and $x^p$ is the random variable representing the label of the node at the p-th step of the traversal. Thus $P(x^p)$ is the probability distribution over the labels at the p-th node. The encoder function, $\text{enc}(S)$, generates a $k$-dimensional encoding of a source tree, $\text{enc} : S \to \mathbb{R}^k$. In general, the encoder does not need to be a tree encoder. However, concretely, we used a TreeLSTM for encoding the input tree $S$.

The decoder for our tree-to-tree transducer can be viewed as a context sensitive grammar. For this reason in sections 2.1.2 and 2.1.3 we introduce the formal notation of the probablistic context free grammar (PCFG) which is the theoretical grounding of our model and relates back to previous literature on tree transduction. However, the reader may skip these sections as the neural tree decoder is specified in section 2.1.4.

### 2.1 NEURAL TREE DECODER

Unlike in sequence learning, which is simply left-to-right or vice versa, tree-based encoding and decoding requires us to select a tree traversal path. Thus, in order to properly define the probability structure over the tree we specify the one-to-one serialization of the binary tree to a sequence. We first start by defining a probabilistic model over trees, which is dependent on the tree traversal (or serialization). Next, we relate the tree decoding to production rules, first using the probabilistic context free grammar, and then we develop the our decoder model which is a probabilistic context sensitive model specification of our neural decoder.

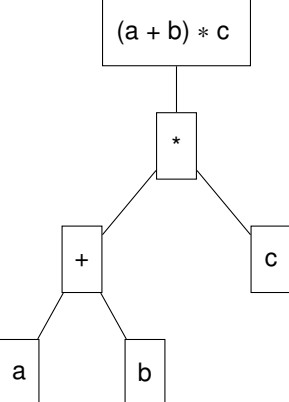

Figure 1: Example binary tree for ( a + b ) * c.

Figure 1 shows a tree encoding for $(a + b) * c$. An example explicit serialization of the tree is " node {*} child{ node {+} child{ node {a}} child{ node {b}} } child{ node {c}} ; ." While this is a

general notation, we can further simplify to serializing the tree as { * { + { a } { b } } { c } } where a subtree is decorated by { }. This tree representation is from a depth first (or left most) derivation. This is also otherwise known as a prefix notation. (One can rewrite the infix notation (a + b) * c into prefix notation * + a b c.) However, in general if an interior node can have only a left or a right child we need minimally use } to decorate the sequence, so * + a } b } } c } } would be a minimal representation.

We require our serial representations to be one-to-one with trees. Thus, a probability distribution over such sequences uniquely corresponds to a probability distribution over trees. There are several ways of serializing trees, which lead to different probability distributions over the resulting strings. Thus, a simple representation for probability over trees requires a good choice for serialization and then a restriction over the probability distribution of these strings.

The derivation path $Path(T)$ is the traversal sequence of the tree. The function $Path(\cdot)$ takes a tree as its argument and outputs a sequence which corresponds to the derivation path. Focusing on the depth first derivation, we can see that the probability of the probability distribution over the p-th output of $x^p$ is conditional on the previous output, so we can write the probability of the next observation $b$ as $P(x^4 = b \mid x^1 = *, x^2 = +, x^3 = a)$.

Thus, we can generalize to
$$P(x^p \mid x^1, \ldots, x^{p-1}).$$
And thus, we can say that the probability of the tree $T$, $P(T)$, is the serialization sequence

$$P(T) = \prod_{p \in Path(T)} P(x^p \mid x^1, \ldots, x^{p-1}) \equiv \prod_{p=1}^{Path(T)} P_{p-1}(x^p).$$

Mapping from one tree to another constitutes a grammar and to understand how our neural tree transducer works we review for context sensitive and context free grammar transduction works. In order to serialize the tree, a set of production rules is required. Thus, in the following sections we describe a formal model of serialization using production rules, starting with the simplest model, context free production rules. We then generalize these context free grammars to make them context sensitive by taking into account the left sibling subtree when producing the right sibling.

### 2.1.1 Tree Production Rules

It is convenient to represent trees as a set of production rules. In particular we can write $S \rightarrow O \mid v \mid \epsilon, O \rightarrow OO \mid vO \mid Ov \mid vv$. In this notation, $S$ is the root node of the tree, $O$ is the set of non-terminals, and $v$ are terminal symbols $\Sigma$ and $\epsilon$. For clarity, in our example we define the non-terminals $O \in \{+, *\}$ and the terminals $v \in \{a, b, c, \epsilon\}$.

The observation is generated by the sequence of left most production rules

$S \rightarrow *$ by $S \rightarrow O$, $* \rightarrow +c$ by $O \rightarrow Ov$, and $+c \rightarrow abc$ by $O \rightarrow vv$

giving a tree which corresponds to figure 1. Production rules are, in general, not adequate for doing serialization since they may be ambiguous; there may be more than one tree corresponding to a resulting sequence. However, production rules are a good language to describe simple probabilistic models for trees. So we need to relate a serialization to production rules. We will first do this for probabilistic context free grammars and then extend to context sensitive ones.

A first step in making this connection between production rules and trees is to define a function $U$ : sequences $\rightarrow$ symbols. When given a partial sequence of a tree, $U$ returns the symbol for the parent of the next symbol to be generated. Next we define a function $V : (\Sigma \cup O) \rightarrow \mathbb{R}^k$ to access a probability vector given a symbol. This allows us to get at the idea of a production function by making claims like, $P(x^p = x \mid x^1, \ldots, x^{p-1}) = P(x^p = x \mid U(x^1, \ldots, x^{p-1})) = g(x \mid u) = g_2(x \mid V(u))$ where $u = U(x^1, \ldots, x^{p-1})$. $U$ then gives us access to the same information as the rule that was used to land us at this point in the tree.

### 2.1.2 Probabilistic Context Free Grammar

Although the above grammar is not, in general, context free, we can make is so by simplifying the assumptions about the production rules. This naturally will limit the conditional probability of a

next observation conditional on the probability of the production rule. In a probabilistic context free grammar (PCFG), all production rules are independent. In other words $P(x^p = x \mid x^1, \ldots, x^{p-1}) = P(x^p = x \mid U(x^1, \ldots, x^{p-1}) = y) = P(y \to x)$. In order to make our grammar context-free, we split and restrict the production rule $O \to OO \mid vO \mid Ov \mid v$ into two productions of $O \xrightarrow[left]{} O \mid v, O \xrightarrow[right]{} O \mid v$.

There are other representations for context-free grammars, which are equivalent, but for our purposes this is the most relevant. For example, $+ \to ab$ becomes $+ \xrightarrow[left]{} a$ and $+ \xrightarrow[right]{} b$. Going back to the example of the probability of the next observation $b$ as,

$$P(x^4 = b \mid x^1, x^2, x^3) = P(x^4 = b \mid U(x^1, x^2, x^3)) = P(x^4 = b \mid +) = P(+ \xrightarrow[right]{} b).$$

The generalization of this model of the probability of a tree is

$$P(T) = \prod_{p \in Path(T)} P(x^p \mid x^1, \ldots, x^{p-1}) = \prod_{p \in Path(T)} P(x^p \mid U(x^1, \ldots, x^{p-1}) = y) = \prod_{p \in Path(T)} P(y \to x^p).$$

### 2.1.3 Probabilistic Context Sensitive Model

We do not, in fact, want to have the production rules be context free, so we will split the production rules $O \to OO \mid vO \mid Ov \mid vv$, in order not to assume independence on the paths, aside from the parent. Given left-most derivations, the production rules can be modified into $O \xrightarrow[left]{} v \mid O$ and $O \xrightarrow[right]{\text{conditioned on left}} OO \mid vO \mid Ov \mid vv$. The left dependency is an assumption of our model; however, our decoder can be easily extensible to other context sensitive assumptions.

For representation purposes, we need to define functions for the derivation path, as well as the left subtree. We define the function $f$, which takes an index $p \in Path(T)$ and returns the index of the parent node. Then we can define the derivation path to $x^p$ as the sequence $\{x^1, \ldots, x^{f(f(p))}, x^{f(p)}\} \equiv \mathcal{P}_p$. Furthermore, we can define a function $\mathcal{LS}$ which extracts the left subtree sequence for an entry $x^p$, so $\mathcal{LS}_p = \{x^j \text{ s.t. } f(p) < j < p\}$.

Having defined the required functions we, can now write down our context sensitive model as $P(x^p \mid x^1, \ldots, x^{p-1}) = P(x^p \mid \{\mathcal{P}_p\} \cup \{\mathcal{LS}_p\})$. Relative to a PCFG, where we have $P(x^p \mid x^{f(p)})$, the context sensitive model relies on the derivation path as well as the left subtree.

Now we define two functions $G : \mathcal{P}_p \to \mathbb{R}^k$ and some $H : \mathcal{LS}_p \to \mathbb{R}^k$. Under our model

$$P(x^p = x \mid x^1, \ldots, x^{p-1}) = P(x^p \mid \{\mathcal{P}_p\} \cup \{\mathcal{LS}_p\}) = P(x^p = x \mid G(\mathcal{P}_p), H(\mathcal{LS}_p)) = g(x \mid h^{\mathcal{P}}, h^{\mathcal{LS}}).$$

Having fully specified the general context sensitive model, we now define the specification of the decoder for our neural tree transducer (NTT).

### 2.1.4 Neural Model Specification

We use can use any recurrent neural network model for representing the information passed down from the root to the p-th node $G$, and for the encoding of the left subtree, $H$ we can use any recursive neural network model.

The recurrent neural network, $G$, and the recursive neural network, $H$, could be specified as

$$h^{\mathcal{P}} = \tanh(V^{\mathcal{P}} h^{f(p)} + U^{\mathcal{P}} x^{f(p)} + b^{\mathcal{P}})$$
$$h^{\mathcal{LS}} = \tanh(V^{\mathcal{LS}_l} h^{C_l((f(p)+1))} + V^{\mathcal{LS}_r} h^{C_r((f(p)+1))} + U^{\mathcal{LS}} x^{f(p)+1} + b^{\mathcal{LS}}),$$

where $C_l$ and $C_r$ are functions which return the left and right direct children respectively, and $x^{f(p)+1}$ is the left sibling of $x^p$. As previously defined $\mathcal{LS}_p$ extracts the left subtree sequence for an entry $x^p$. If $x^p$ is a left child or the root then $h^{\mathcal{LS}} = 0$. The matrices $V^{\mathcal{P}}, U^{\mathcal{P}}, V^{\mathcal{LS}_l}, V^{\mathcal{LS}_r}, U^{\mathcal{LS}}$ are estimated, and differ from the functions $U$ and $V$.

In this paper, an LSTM and a binary TreeLSTM were used for $G$ and $H$, respectively.

Finally, we can write $g(x|h^{\mathcal{P}}, h^{\mathcal{LS}}) = softmax(W^{\mathcal{P}}h^{\mathcal{P}} + W^{\mathcal{LS}}h^{\mathcal{LS}} + b)$. As a direct result,

$$P(x^p = x_i|x^1, \ldots, x^{p-1}) = P(x^p = x_i \mid G(\mathcal{P}_p), H(\mathcal{LS}_p)) = \frac{g(x_i \mid h^{\mathcal{P}}, h^{\mathcal{LS}})}{\sum_{j=1}^{|V|} g(x_j \mid h^{\mathcal{P}}, h^{\mathcal{LS}})},$$

where $x_i$ and $x_j$ represent symbols in the vocabulary, and $|V|$ is the number of symbols in the vocabulary. Figure 2 shows a concrete example of the decoder.

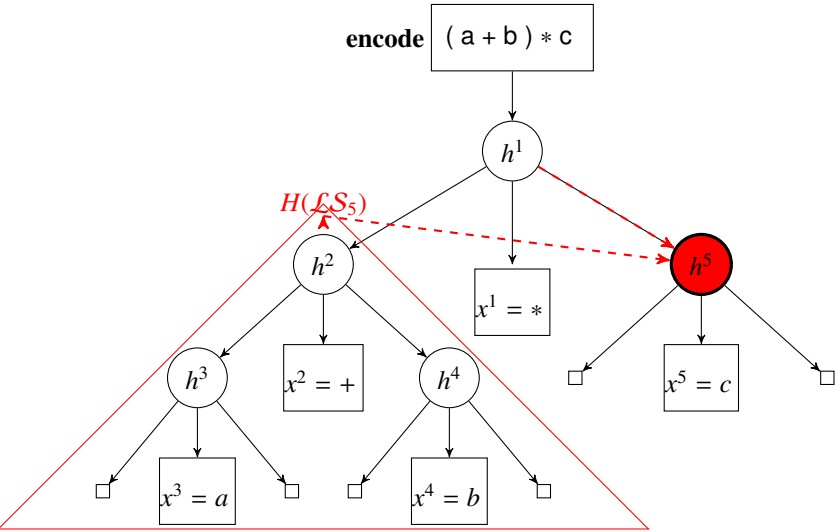

Figure 2: This is an explicit example of the neural tree transducer model (NTT) binary tree for ( a + b ) * c. $x^p$ are the random variables, and $h^p$ are the hidden states. $p$ corresponds to the derivation number, so $x^2$ is the second generated random variable, who's realization is the symbol +. The set $\mathcal{LS}_5 = \{x^2, x^3, x^4\}$, and $\mathcal{P}_5 = \{x_1\}$ The hidden state $h^5$ is a function of the encoded left subtree $h^{\mathcal{LS}_5} = H(\mathcal{LS}_5)) \neq h^2$ and the parent hidden state, $h^{\mathcal{P}} = h^1$. The reader should note that the predicted $\Pr(x^2 \mid x^1) = g(x^2 = + \mid h^2)$, while $\Pr(x^5 \mid x^1, x^2, x^3, x^4) = g(x^5 = c \mid h^2, h^{\mathcal{LS}_5})$, so the difference in the prediction of $x^2$ and $x^5$ is due to the dependence on the left sibling subtree. Finally, $h^1 = enc(S)$ where $S$ is the source tree to be transduced.

An important assumption of the model is that left derivations depend only on their parents, whereas right derivations are sensitive to the left sibling subtree. Allowing the left derivations to be dependant on their right subtree would create a cyclical dependency. Therefore, we make the implicit assumption that the parent hidden state captures sufficient information. In contrast, a breadth first approach can be bidirectionally dependent; however, the independence assumption is made with the children at subsequent depths. Although in this paper we focus on binary trees, we can generalize to N-ary trees by adding a symbol to our vocabulary which does not allow another right sibling.

## 3 RELATED WORK

There are three main related neural methods: Seq2Seq, Seq2Tree and Tree2Seq. To our knowledge there are no explicit Tree2Tree neural models to compare against. As discussed below, Tree2Tree is a transduction problem with is often done implicitly by these methods by using serialized versions of trees. By focusing purely on the tree transduction problem, we obtain both higher accuracy due to better localization and insight into the models. Seq2Seq is a baseline as Seq2Seq can be decomposed into Seq2Tree, Tree2Tree, and Tree2Seq. Thus Seq2Seq models should in theory be able to capture tree transduction.

### 3.1 SEQUENCE-TO-SEQUENCE

The first neural sequence-to-tree method treated grammar as a foreign language (Vinyals et al., 2015a). However, similar to our baseline, their tree output is a serialization of a tree rather than a

tree structure. This is an important distinction from a true Seq2Tree in that the hidden state vector in the decoder must treat the serialization as a stack. In our results, one can see that which serialization one uses matters. Seq2Seq has a disadvantage in that the trees generated may not be well formed. We find this is actually a problem in practice, not just in theory. Dyer et al. (2016) introduced the stack-based RNNG model, which uses a stack structure to create well formed trees. However, this model is still fundamentally a Seq2Seq approach with the stack function serving as a similar form of compression. Similarly, Grefenstette et al. (2015) show that the augmentation of Seq2Seq with memory was required for higher transduction accuracy.

## 3.2 Tree-to-Sequence

Tree-to-Sequence learning can be thought of as a tree serialization task (Eriguchi et al., 2016b;a). However, the tree representation is also useful for localizing information. Furthermore, as (Socher et al., 2011) showed, tree-to-value clearly captures sentiment. The idea that locality is more accurately captured in trees seems natural and could plausibly replace the (somewhat *ad hoc*) input reversal in Seq2Seq. Chen et al. (2017) showed improvements in machine translation by incorporating a tree encoder; however, their decoder remains sequence based.

## 3.3 Sequence-to-Tree

Sequence-to-Tree transformation is a more well-studied problem, since it includes parsing and neural parsing has achieved state-of-the-art results (Aharoni & Goldberg, 2017) and hence garnered much attention. There are two main differences between these methods and our approach, 1) other decoders are breadth based, and 2) many sequence-to-tree problems, particularly dependency parsing, are fundamentally not tree-to-tree.

Recently, several neural tree decoders have been proposed. The models most similar to our approach are the SEQ2TREE (Dong & Lapata, 2016), top-down tree LSTMs (Zhang et al., 2015), and the double recurrent neural networks (Alvarez-Melis & Jaakkola, 2017). All of their neural decoders, while somewhat similar to our approach, differ in important ways from our decoder. Most importantly, all of these methods use top down breadth first search, whereas our model is depth first.

Dong & Lapata (2016) used stack based production rules using a hierarchical decoder; however, SEQ2TREE does not explicitly do context sensitive production. Top-down tree LSTMs (Zhang et al., 2015) have four independent LSTMs for decoding, which require expensive explicit tree serialization and thus more parameters than we use. While doubly recurrent NNs (Alvarez-Melis & Jaakkola, 2017) do not suffer from the issue of multiple independent LSTMs, it does require explicit differentiation predictions of both terminal nodes and subsequently node labels.

Direct comparison of our tree-to-tree method on the problems addressed in these seq-to-tree papers is not possible. Dependency parsing, as done in (Zhang et al., 2015) is inherently sequence-to-tree, and does not map cleanly onto any tree-to-tree problem. Mapping sentences to functional programs, as done in (Alvarez-Melis & Jaakkola, 2017; Dong & Lapata, 2016) could be made into a tree-to-tree problem by parsing the sentences into trees, but their approach uses explicitly constructed n-ary trees where each level is restricted to a given label type.

Our depth first search allows us, at the cost of maintaining a stack, to avoid the problem of the separation of the point at which the label and the child node is predicted, which drove the need for double recurrence in the prior work.

## 4 Experiments

We evaluated the different methods using tree edit distance (TED) (Tai, 1979; Bille, 2005) between the predicted tree and the actual tree. TED is the number of insertions, deletions, and modifications to make the two trees equivalent, while aligning the trees. In order to assess the tree-to-tree transduction rules which the models could learn, we chose four standard tree tasks 1) tree copy, 2) tree reordering, 3) node relabeling, and 4) node and subtree deletion. Our experiments are similar to those of Grefenstette et al. (2015); however, we explicitly isolate complex tree reordering, subtree deletion, and context sensitive relabeling.

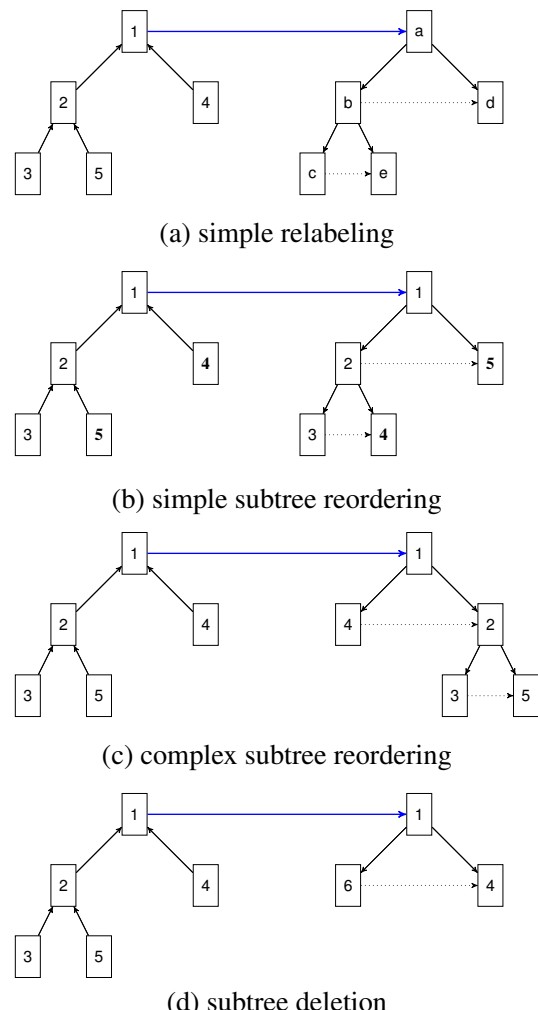

(a) simple relabeling

(b) simple subtree reordering

(c) complex subtree reordering

(d) subtree deletion

Figure 3: a) Simple relabeling example where $1 \rightarrow a$, $2 \rightarrow b$, and so on. b) subtree reordering where nodes labeled 4 and 5 swap. c) full reordering of subtree where left and right subtrees of the root node swap, and d) subtree deletion where the subtree of node 2 is replaced be the node labelled 6. The dotted arrows represent conditional dependence.

**Tree Copy and Simple Node Relabeling** While tree copying seems to be trivial, it shows that both Seq2Seq and Tree2Tree can perform copies. The NTT does not reverse order, but instead encodes the tree bottom up and left to right. One interesting observation is that Tree2Tree training required more training examples.

For tree copying, we simply validated that all of the models could copy a tree. For simple relabeling, we used context free relabeling rules to change from one vocabulary to another, so $1 \rightarrow a$. (See Figure 3.)

**Complex Node Relabeling** For complex relabeling, or context sensitive relabeling, we took a tree with arbitrary labels and transformed it into a tree where the labels are reordered. We tried two subtasks, the first where the left sibling was required to be smaller than the right sibling. The second, more complex, task required the right sibling to be less than the left subtree. This causes the tree to be ordered.

**Tree Reordering** For tree reordering, we generated examples by taking mathematical expressions in reverse polish (or prefix notation) and transforming them infix notation. This process is a non-trivial

tree reordering. However, there is no node relabeling, so it isolates the reordering capability of the NTT model. As expected, this is where the most benefit of the NTT model was found.

**Subtree deletion** For our final experiment, we focused on deletion rules. Here we created transduction rules to delete a subtree and replace it with a new symbol.

## 4.1 SETTINGS

Throughout our experiments we chose a maximum tree depth of 6 and vocabulary size of 256.

For our baseline Seq2Seq model, we used a two layer LSTM with 256 dimensions for both the hidden units and the memory cell with a bidirectional encoder and we also used the same formulation for the Seq2Seq with attention. Our Neural Tree Transducer (NTT) used a binary TreeLSTM for the encoder with a 256 dimensional hidden state space. The left subtree encoder was also a binary TreeLSTM, but with only one layer and the parent derivation was a LSTM with one hidden layer with 256 hidden units.

We used teacher forcing and curriculum learning. For all models we used the AdamOptimizer with an initial learning rate of 0.001, a dropout ratio of 0.2 and gradient clipping at 5.0. The models were trained until the development accuracy no longer improved.

## 4.2 RESULTS

Results are shown in Table 1, and discussed below. For the task of simple relabeling, both Seq2Seq-Attn and NTT models perform nearly perfectly, which shows (not surprisingly) that the simple relabeling is not difficult. The Seq2Seq model without attention uniformly performs worse; the attention mechanism greatly aids relabeling. However, for complex relabeling the NTT performs much better than the Seq2Seq attention model. This is to be expected, as context sensitive relabeling relies heavily on the left subtree. Furthermore, the Seq2Seq model performs only sightly above the no edit case where the prediction is the original tree. In comparing the reordering, the NTT also greatly outperforms the baselines. Again, this is expected as the localization of a reordering is much easier in the NTT model. Finally, for deletion the NTT does not largely outperform Seq2Seq with attention model. The deletion is localized to a subsequence of the serialization. As a result, the attention model seems to be able to accurately "forget", while for the NTT model this is a equivalent to a relabeling task.

| Task | Method | Tree Depth | | | |
|---|---|---|---|---|---|
| | | 3 | 4 | 5 | 6 |
| Simple Relabeling | Seq2Seq | 0.04 | 0.12 | 0.98 | 2.43 |
| | Seq2Seq-Attn | **0** | **0** | 0.36 | 0.72 |
| | NTT | **0** | **0** | **0** | **0.56** |
| Complex Relabeling | Seq2Seq | 0.55 | 1.40 | 3.40 | 9.62 |
| | Seq2Seq-Attn | 0.27 | 1.05 | 2.32 | 6.03 |
| | NTT | **0.17** | **0.41** | **0.96** | **3.76** |
| Reordering | Seq2Seq | 0.25 | 1.69 | 4.33 | 13.47 |
| | Seq2Seq-Attn | 0.21 | 1.26 | 3.36 | 10.93 |
| | NTT | **0.11** | **1.07** | **2.15** | **5.01** |
| Deletion | Seq2Seq | 0.15 | 0.61 | 1.44 | 6.23 |
| | Seq2Seq-Attn | **0.13** | 0.39 | 0.81 | 2.57 |
| | NTT | **0.13** | **0.38** | **0.72** | **2.32** |

Table 1: Tree edit distance for the tree tasks of tree copying (Copy), subtree reordering (Reordering), node relabeling (Relabeling) and node and subtree deletion (Deletion). The models are sequence-to-sequence (Sutskever et al., 2014) (Seq2Seq), sequence-to-sequence with attention (Bahdanau et al., 2014) (Seq2Seq-Attn), and our neural tree transducer method (NTT).

## 5 Conclusion and Future Work

We introduced a neural tree transduction (NTT) model which learns Tree2Tree transformations and presented a novel and intuitive context sensitive depth first decoder based on context sensitive grammars. Our NTT model was evaluated over several standard tree tasks and showed superior performance relative to Seq2Seq models. For tree transduction tasks such as syntactic machine translation, tree code based optimization, and text simplification, tree-to-tree is an essential component which is not easily captured by Seq2Seq models.

Unlike top down breadth first decoders, our NTT method naturally lends itself to streaming decoding (Alur & D'Antoni, 2012). Streaming decoding would allow for arbitrary depth decoding and a parallel encoding/decoding scheme. Another immediate improvement to our model would be to incorporate tree attention as in the work of Munkhdalai & Yu (2016), which have applied attention to tree alignment. Finally, we intend to apply our method to syntactic machine translation (translating from a dependency parse in one language to the parse in another language), text simplification, and program optimization.

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

Xingxing Zhang, Liang Lu, and Mirella Lapata. Top-down tree long Short-Term memory networks. 31 October 2015.

