# OpenReview forum: "Neural Tree Transducers for Tree to Tree Learning"
_ICLR.cc/2018/Conference — Reject_

### Official Review · AnonReviewer2 · 2017-11-27
**Very limited contribution**

**Rating:** 3
**Confidence:** 4

**Review:**

The paper introduces a neural tree decoder architecture for binary trees that conditions the next node prediction on
representations of its ascendants (encoded with an LSTM recurrent net) and left sibling subtree (encoded with a binary LSTM recursive net) for right sibling nodes.
To perform tree to tree transduction the input tree is encoded as a vector with a Tree LSTM; correspondences between input and output subtrees are not modelled directly (using e.g. attention) as is done in traditional tree transducers.
While the term context-sensitive should be used with caution, I do accept the claim here, although the notation used does not make the exposition clear.
Experimental results show that the architecture performs better at synthetic tree transduction tasks (relabeling, reordering, deletion) than sequence-to-sequence baselines.

While neural approches to tree-to-tree transduction is an understudied problem, the contributions of this paper are very narrow and it is not shown that the proposed approach will generalize to more expressive models or real-world applications of tree-to-tree transduction.
Existing neural tree decoders, such as Dong and Lapata or Alvarex-Melis and Jaakkola, could be combined with tree LSTM encoders without any technical innovations and could possibly do as well as the proposed model for the transduction tasks tested - no experiments are performed with existing tree-based decoder architectures.

Specific comments per section:

1. Unclear what is meant be "equivalent" in first paragraph.
2. The model does not assign an explicit probability to the tree structure - rather it seems to rely on the distinction between terminal and non-terimal symbols and the restriction to binary trees to know when closing brackets are implied - this is not made clear, and a general model should not have this restriction, as there are many cases where we want to generate non-binary trees.
The production rule notation used is incorrect and confusing, mixing sets with non-terminals and terminal symbols:
A better notation for the rules in 2.1.1 would be something like S -> P | v | \epsilon; P -> Q R | Q u | u Q | u w, where P, Q, R \in O and u, w \in v.
2.1.2. Splitting production rules as ->_left, ->_right is not standard notation. Rather introduce intermediate non-terminals in the grammar:
O -> O_L O_R; O_L -> a | Q, O_R -> b | Q.
2.1.3 The context-sensitively here arise when conditioning on the entire left sibling subtree (not just the top non-terimal).
The rules should have a format such as O -> O_L O_R; O_L -> a | Q; \alpha O_R -> \alpha a | \alpha Q, where \alpha is an entire subtree rooted at O_L.
2.1.4 Should be g(x|.) = exp( ), the softmax function includes the normalization which is done in the equation below.

3. Note that is is possible to restrict the decoder to produce tree structures while keeping a sequential neural architecture. For some tasks sequential decoders do actually produce mostly well-formed trees, given enough training data.
RNNG encodes completed subtrees recursively, and the stack LSTM encodes the entire partially-produced tree, so it does produce and condition on trees not just sequences. The model in this paper is not more expressive than RNNG, it just encodes somewhat different structural biases, which might or might not be suited for real tasks.

4. In the examples given, the same set of symbols are used as both terminals and non-terminals. How is the tree structure then predicted by the decoder?
Details about the training setup are missing: How is the training data generated, what is the size of the trees during training (compared to testing)?
4.2 The steep drop in performance between depth 5 and 6 indicates that model is very sensitive to its memorization capacity and might not be generalizing over the given training data.
For real tree-to-tree applications involving these operations, there is good reason to believe that some kind of attention mechanism will be needed over the input tree during decoding.

Reference should generally be to published proceedings rather than to arxiv where available - e.g. Aharoni and Goldberg, Dong and Lapata, Erguchi et al, Rush et al. For Graehl and Knight there is a published journal paper in Computational Linguistics.

---

### Official Review · AnonReviewer1 · 2017-12-01
**Would benefit from a better evaluation setup.**

**Rating:** 7
**Confidence:** 4

**Review:**

The authors propose to tackle the tree transduction learning problem using recursive NN architectures: the prediction of a node label is conditioned on the ancestors sequence and the nodes in the left sibling subtree  (in a serialized order)
Pros:
- they identify the issue of locality as important (sequential serialization distorts locality) and they move the architecture closer to the tree structure of the problem
- the architecture proposed moves the bar forward in the tree processing field
Cons:
- there is still a serialization step (depth first) that can potentially create sharp dips to null probabilities for marginal changes in the conditioning sequence (the issue is not addressed or commented by the authors)
- the experimental setup lacks a perturbation test: rather than a copy task, it would be of greater interest to assess the capacity to recover from noise in the labels (as the noise magnitude increases)
- a clearer and more articulated comparison of the pros/cons w.r.t. competitive architectures would improve the quality of the work: what are the properties (depth, vocabulary size, complexity of the underlying generative process, etc) that are best dealt with by the proposed approach?
- it is not clear if the is the vocabulary size in their model needs to increase exponentially with the tree depth: a crucial vocabulary size  vs performance experiment is missing

---

### Official Review · AnonReviewer3 · 2017-12-04
**There may be some interesting ideas here, but I think in many places the mathematical description is very confusing and/or flawed.**

**Rating:** 2
**Confidence:** 5

**Review:**

There may be some interesting ideas here, but I think in many places the mathematical
description is very confusing and/or flawed. To give some examples:

* Just before section 2.1.1, P(T) = \prod_{p \in Path(T)} ... : it's not clear
at all clear that this defines a valid distribution over trees. There is an
implicit order over the paths in Path(T) that is simply not defined (otherwise
how for x^p could we decide which symbols x^1 ... x^{p-1} to condition
upon?)

* "We can write S -> O | v | \epsilon..." with S, O and v defined as sets.
This is certainly non-standard notation, more explanation is needed.

* "The observation is generated by the sequence of left most
production rules". This appears to be related to the idea of left-most
derivations in context-free grammars. But no discussion is given, and
the writing is again vague/imprecise.

* "Although the above grammar is not, in general, context free" - I'm not
sure what is being referred to here. Are the authors referring to the underlying grammar,
or the lack of independence assumptions in the model? The grammar
is clearly context-free; the lack of independence assumptions is a separate
issue.

* "In a probabilistic context-free grammar (PCFG), all production rules are
independent": this is not an accurate statement, it's not clear what is meant
by production rules being independent. More accurate would be to say that
the choice of rule is conditionally independent of all other information
earlier in the derivation, once the non-terminal being expanded is
conditioned upon.

---

### Decision · Program_Chairs · 2018-01-29
**ICLR 2018 Conference Acceptance Decision**

**Decision:**

Reject

**Comment:**

The proposed neural tree transduction framework is basically a combination of tree encoding and tree decoding. The tree encoding component is simply reused from previous work (TreeLSTM) whereas the decoding components is somewhat different from the previous work. They key problems (acknowledge also by at least 2 reviewers):

Pros:
-- generating trees input under-explored direction (note that it is more general than parsing as nodes may not directly correspond to input symbols)

Cons:
-- no comparison with previous tree-decoding work
-- only artificial experiments
-- the paper is hard too read (confusing) / mathematical notation and terminology is confusing and seems sometimes inaccurate (see R3)